# Unified and Generalizable Reinforcement Learning for Facility Location Problems on Graphs

## Abstract

Facility location problems on graphs are ubiquitous in the real world and hold significant importance, yet their resolution is often impeded by NP-hardness. MIP solvers can find the optimal solutions but fail to handle large instances, while algorithm efficiency has a higher priority in cases of emergency. Recently, machine learning methods have been proposed to tackle such classical problems with fast inference, but they are limited to the myopic constructive pattern and only consider simple cases in Euclidean space. This paper introduces a unified and generalizable approach to tackle facility location problems on weighted graphs with deep reinforcement learning, demonstrating a keen awareness of complex graph structures. Striking a harmonious balance between solution quality and running time, our method stands out with superior efficiency and steady performance. Our model trained on small graphs is highly scalable and consistently generates high-quality solutions, achieving a speedup of more than 2000 times to Gurobi on instances with 1000 nodes. The experiments on Shanghai road networks further demonstrate its practical value in solving real-world problems.

## CCS Concepts

• **Networks → Network design and planning algorithms**; • **Computing methodologies → Reinforcement learning**.

## Keywords

Facility location problems, Graphs and networks, Combinatorial optimization, Deep reinforcement learning

**ACM Reference Format:**
Anonymous Author(s). 2018. Unified and Generalizable Reinforcement Learning for Facility Location Problems on Graphs. In *Proceedings of Make sure to enter the correct conference title from your rights confirmation emai (Conference acronym 'XX)*. ACM, New York, NY, USA, 14 pages. https://doi.org/XXXXXXX.XXXXXXX

## 1 Introduction

Facility location problems (FLPs) study optimizing the placement of a set of facilities to meet customer demands and minimize some objective function. Various models are employed to address specific requirements, including single/multiple facility problems, median location problems, dynamic location problems, etc [4]. According

to Farahani and Hekmatfar [10], there are four essential components defining location problems: customers, facilities to be located, a space in which customers and facilities are located, and a distance metric between customers and facilities. Among the distance metrics used, routing distance in networks is notable, wherein customers and facilities are positioned on the nodes of a graph. To address optimization problems in non-Euclidean spaces, one approach involves computing the pairwise distance matrix between all nodes and reformulating the problem into a general framework. Figure 1 illustrates the process of solving real-world facility location problems on graphs. Initially, real-world networks from diverse domains are abstracted into graphs. The demands at each node and the pairwise distances between nodes are utilized to encode problems as mathematical optimization models. This quantitative representation, along with specific constraints inherent to various problem types, is then input into solvers for final solutions.

Among various concrete problems of facility location problems in networks, we focus on two of them, namely the $p$-median problem (PMP) and the facility relocation problem (FRP). PMP is a crucial branch of facility location problems that seeks to minimize the weighted sum of distance costs between facilities and demand points, with fixed costs for opening facilities and a predefined number of facilities $p$. It finds applications in various fields, including designing electric charging station networks [13], establishing public services such as schools [26], and siting shared bicycles [7]. While the $p$-median model proves effective in static scenarios, it encounters limitations in dynamic and constrained environments. Take urban infrastructure constructed during a city's early development stages. The dynamic and evolving population in this area over time leads to a mismatch between actual demands and the outdated facility layout. Under such circumstances, it becomes more pragmatic to relocate facilities rather than plan anew, especially given economic constraints limiting the number of relocations. Another pertinent example involves the short-term need to rebalance a bicycle-sharing system to adjust to the frequent redistribution of bikes due to user usage patterns. These scenarios give rise to the second problem of interest: improving the existing facility layout within a constrained number of relocation steps, known as the facility relocation problem. For FLPs in case of emergency [19, 32] and scenarios with frequent demand shifts, there are higher requirements for algorithm efficiency and they often prioritize running time over optimality.

Many heuristics and meta-heuristics have been devised to solve FLPs. From the perspective of how the solution evolves as the algorithm progresses, they can be categorized into two genres: constructive methods and improving methods. Constructive methods start with an empty set of facilities and build the solution incrementally. In contrast, improving heuristics aim to enhance a feasible solution through modifications. Constructive methods typically

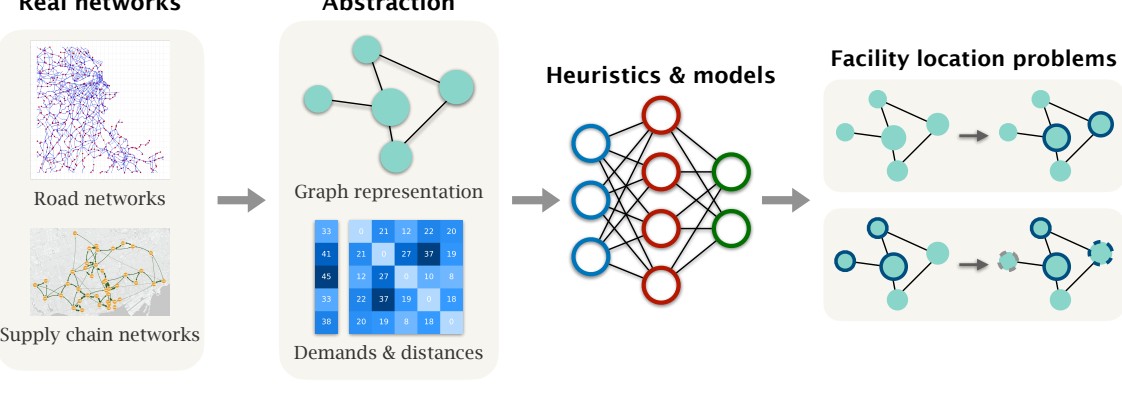

**Figure 1: The general pipeline of solving real-world facility locations problems on graphs. Real-world networks are converted into abstract graph representations. The demands of each node and pairwise distances are used to encode the problems as mathematical optimization models. This quantitative representation along with problem constraints is fed into constraint solvers, yielding the final solutions for various FLPs.**

exhibit myopic behavior by not altering or undoing chosen facilities. Empirical results corroborate this observation, indicating that interchange algorithms achieve lower optimality gaps [16], albeit at the expense of increased runtime.

The recent advancement in machine learning, particularly deep learning, offers an alternative perspective on solving these classical problems. Compared to general-purpose constraint programming solvers, the strong expressiveness and rapid inference capabilities of neural networks make them powerful tools for tackling complex combinatorial optimization problems [2]. This is particularly advantageous in times of emergency that demand real-time responses to large-scale problems. However, previous works in this field predominantly follow the constructive approach of creating solutions [18, 21, 23, 30]. Moreover, previous studies are limited to a simplified geometrical setting without considering graph structures. In many scenarios, such as urban planning and network routing, graph distances can portray actual traveling costs more accurately than straight-line distance [29].

To address these limitations, we propose a highly scalable swap-based approach to solve facility location problems in combination with reinforcement learning. Our model demonstrates a keen awareness of the complex graph structures of the instances and the solving states, enabling it to make improvement decisions effectively and efficiently. Experiments reveal that our improving-style method demonstrates stronger generalizability compared to constructive-style peer methods with deep learning.

In conclusion, the main contributions of this paper are as follows:

- Unified approach: We introduce a unified algorithm capable of simultaneously solving two FLPs, i.e. the relocation problem and the $p$-median problem.
- Generalizability: The novel improving-style algorithm showcases superior generalizability across varying graph sizes and facility numbers. It has more steady performance and is less sensitive to instance parameters.
- Performance and scalability: Our method makes rapid responses to large instances up to thousands of nodes within

seconds, yielding high-quality solutions and making significant acceleration to peer methods.
- Complexity handling: Through delicately designed features, our model can handle FLPs with complex structures and graph distances.

## 2 Related Work

We consider facility location problems in discrete solution spaces, categorizing them as combinatorial optimization (CO) problems. The survey [2] provides a comprehensive review of the intersection of machine learning and combinatorial optimization.

### 2.1 Machine Learning for Facility Location Problems

Several works have explored solving PMP with machine learning techniques. Most of these approaches formulate the solution construction scheme as a Markov decision process and build solutions step-by-step. For the $p$-median problem, Wang et al. [30] first propose to solve the uncapacitated $p$-median problem in the Euclidean space with reinforcement learning and graph attention networks. They use the REINFORCE [33] algorithm to choose the next facility in the solution. Zhao et al. [36] use DQN [24] to address the capacitated $p$-median problem. Matis and Tarábek [23] solve the weighted $p$-median problem with reinforcement learning and convolutional neural networks. [18] is a recent work that approaches a line of spatial optimization problems with an encoder–decoder structure called SpoNet. The above formulation has been adapted to other FLPs, including the maximal covering location problem (MCLP) [28], $p$-center problem (PC) [6], etc. As for FRP, Luo et al. [21] address the facility relocation problem with a twofold objective of facility exposure and user convenience. They use a reinforcement learning module as an assistive component to a greedy algorithm that maximizes the single-step reward.

Our work is the first machine learning method that addresses the FLPs from an improving perspective and handles complex graph structures of instances. Our agent exhibits a much higher level

of autonomy compared to [21], as it can choose which facility to relocate and its destination.

## 2.2 Machine Learning for Solution Improvement

Though most machine learning solutions to CO problems build solutions incrementally, there are some works exploring general improvement-style algorithms in a broader scope. The pioneering work of Chen and Tian [5] introduces NeuRewriter, a reinforcement learning model that learns region-picking and rewriting-rule policies. This model is applied to expression simplification, job scheduling, and capacitated vehicle routing problems. In Lu et al. [20], the focus is on enhancing solutions to the capacitated vehicle routing problem, incorporating perturbation operators for a larger search space. Additionally, Wu et al. [34] consider improving heuristics for two routing problems using a compatibility layer computed based on query and key from self-attention layers. Summarizing three intervention points of meta-heuristics, Falkner et al. [9] design a policy model based on graph neural networks to assist local search, conducting experiments on job shop scheduling and capacitated vehicle routing problems. Garmendia et al. [12] combine graph neural networks with hill-climbing-based algorithms to improve solutions for preference ranking problem, traveling salesman problem, and the graph partitioning problem. Zhang et al. [35] propose a RL-guided improvement heuristic for solving job-shop scheduling problems. It's noteworthy that most works in this domain primarily focus on routing problems, which are sensitive to the sequential order of nodes, with objective functions solely defined by adjacent nodes in the solution sequence. The different problem structure of FLP introduces more complexity and poses unique challenges.

## 3 Preliminaries and Formulation

In this section, we formally define two typical types of FLPs on undirected weighted graphs.

### 3.1 P-median Problem

We study the $p$-median problem defined on a graph $G(V, E)$, given coordinates $(x_i, y_i)$ and demand $p_i$ for each node $i \in V$. The edges $E$ represent available routes for traveling between nodes, and the traveling costs are determined by the lengths of the shortest paths rather than straight-line distances. Assuming each facility possesses infinite capacity and one node can only accommodate one facility, the objective of PMP on $G$ is to select a facility set $F \subseteq V$ of the predefined size $p$ to minimize the overall traveling cost. This cost is defined as the weighted sum of costs from nodes to their nearest facilities. Let $n = |V|$ denote the number of nodes. The traveling cost between nodes is defined by the shortest paths on $G$, expressed through the distance matrix $D \in \mathbb{R}^{+n \times n}$, where $d_{ij}$ signifies the distance between nodes $i$ and $j$. The distance matrix can be conveniently computed offline using Dijkstra's algorithm Dijkstra [8]. Formally, the objective function $O(F)$ and the optimal facility set

**Algorithm 1:** A general swap framework for facility relocation problem

---

**Parameters:** iteration number $T$, swapping model *agent*
**Input:** graph $G$, existing facilities $F_0$, relocation budget $k$
**Output:** relocation pair $(R_k, I_k)$, new facility set $F$

1 **Function** SwapRelocate($T, agent, G, F_0, k$)
2     $R_k, I_k, F_k \leftarrow \emptyset, \emptyset, F_0$;
3     **for** $i \leftarrow 1$ **to** $T$ **do**
4        $R, I, F \leftarrow \emptyset, \emptyset, F_0$;     // removed, inserted & current facilities
5        **for** $j \leftarrow 1$ **to** $k$ **do**
6           $(u_1, u_2) \leftarrow agent.\text{act}(G, F)$;
7           $R, I, F \leftarrow R \cup \{u_1\}, I \cup \{u_2\}, F \setminus \{u_1\} \cup \{u_2\}$;
8           **if** $O(F) < O(F_k)$ **then**
9              $R_k, I_k, F_k \leftarrow R, I, F$;
10     **return** $(R_k, I_k), F$;

---

$F^*$ are articulated as follows:

$$O(F) = \sum_{i \in V} p_i \min_{j \in F} d_{ij}, \quad \text{s.t. } F \subseteq V, |F| = p. \tag{1}$$

$$F^* = \arg \min_{F} O(F). \tag{2}$$

### 3.2 Facility Relocation Problem

Different from the classical $p$-median model, the facility relocation problem considers a dynamic demand changed over time and the facilities should be relocated correspondingly to meet people's needs. For example, in urban areas, the population density may shift due to new residential developments or changes in public transportation routes, leading to varying demands. For a predefined set of facilities $F_0 \subset V$ and a limited budget $k$, we study the improvement achieved by moving at most $k \leq |F_0|$ facilities within $F_0$. This relocation is represented by a pair of sets $(R_k, I_k)$. The updated facility set is defined as $F = F_0 \cup I_k \setminus R_k$. The fundamental assumptions regarding traveling costs and demands remain consistent with the $p$-median model. Formally, the objective of relocation problem $O_k(R_k, I_k|F_0)$ is defined based on (1):

$$O_k(R_k, I_k|F_0) = O(F_0 \cup I_k \setminus R_k), \tag{3}$$

$$\text{s.t.} \quad R_k \subseteq F_0, \ I_k \subseteq V \setminus F_0, \ |R_k| = |I_k| \leq k. \tag{4}$$

We further define the improvement ratio $Q$ of a relocation set pair as the ratio of decreased cost to original cost before relocation:

$$Q(R_k, I_k|F_0) = \frac{O(F_0) - O_k(R_k, I_k|F_0)}{O(F_0)}. \tag{5}$$

## 4 Methods

We start with a general swap-based framework for solving the facility relocation problem, as shown in Algorithm 1. This framework presents the high-level logic of an improving-style algorithm. Given the set of existing facilities $F_0$ and the maximum number of relocation $k$, it incrementally builds the relocation set pair with the instructions of the given agent. The agent selects a removed facility $u_1$ and an inserted facility $u_2$ for $k$ iterations, and the best

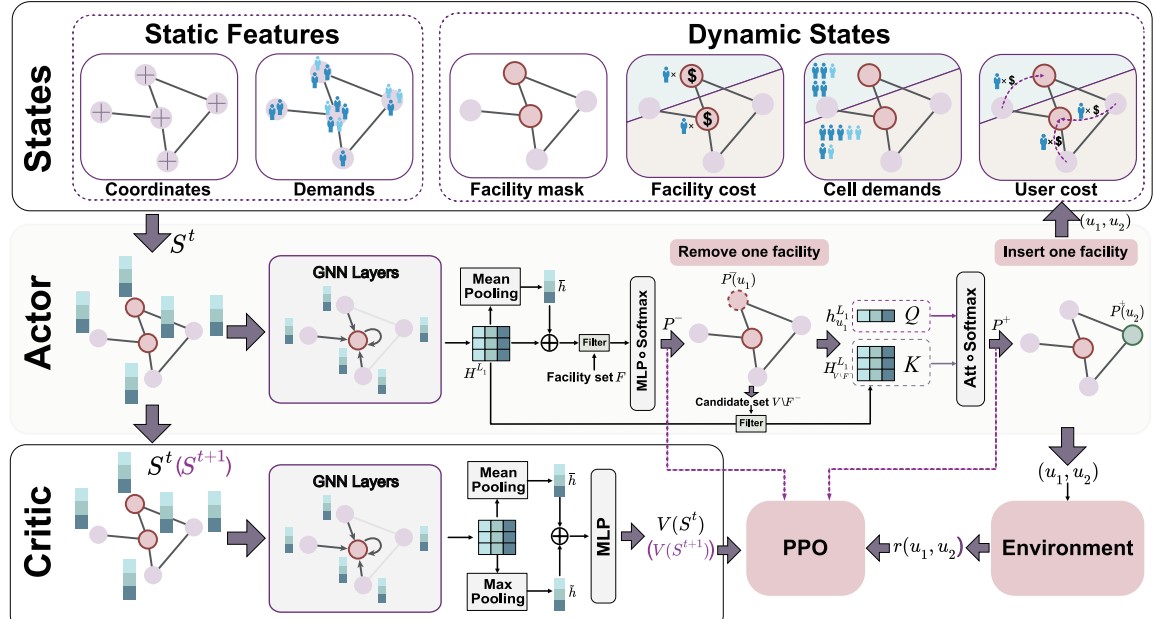

**Figure 2: The model architecture of PPO-swap for solving facility location problems. Node features include static attributes and dynamic states of solutions. We use graph neural networks (GNN) for node embeddings and integrate attention layers to generate two-step actions. The critic evaluates the states based on the global graph context. The training process is powered by the PPO algorithm.**

solution is updated if a lower objective is achieved. This framework is flexible and compatible with various algorithms, including classical handcrafted heuristics. Apart from solving FRP independently, Algorithm 1 also serves as a functional module for solving PMP from scratch, as discussed in Section 4.5.

## 4.1 Reinforcement Learning Formulation for Facility Relocation

We model the relocation pair selection as a Markov decision process, with key components of reinforcement learning defined as follows. The states encompass static attributes of the instances, such as node coordinates and demands, along with the dynamic statistics associated with the current feasible solution (see Section 4.3 for details). Let $R$ be the current set of removed facilities and $I$ the inserted facilities during exploration (consistent with Algorithm 1). The action space is $F \times (V \setminus F)$, one facility to be removed and one to be inserted. Let $(u_1^i, u_2^i)$ be the relocation pair at the $i$-th step. The corresponding reward $r_i$ is defined as the improvement ratio gained from step $i$, namely

$$r_i(u_1^i, u_2^i) = Q(R \cup \{u_1^i\}, I \cup \{u_2^i\}|F_0) - Q(R, I|F_0). \quad (6)$$

## 4.2 PPO-swap: a Learning-based Interchange Algorithm

As an instantiation of Algorithm 1, we employ the proximal policy optimization (PPO) algorithm [27] to train an intelligent agent named **PPO-swap**, designed to learn relocation strategies. PPO-swap adopts an actor-critic network architecture. The actor consists

of $L_1$ graph convolutional layers for node embedding, $L_2$ multi-layer perceptrons (MLP) layers for scoring, and an extra attention layer to choose node pairs conditionally. The critic has a similar structure except for the last attention layer. The overall architecture is depicted in Figure 2. The integration of PPO and GNN equips our model to extract crucial features from complex graph structures, empowering it to make effective decisions.

## 4.3 Voronoi-aware Graph Feature Extractor

We now introduce the node features and graph embeddings of our model, used in both actor and critic. For an input problem instance, the basic attributes of nodes remain unchanged during the solving process, including node coordinates $(x_i, y_i)$ and node demand $p_i$. Dynamic features of the states are crucial for identifying beneficial relocation pairs since the improving algorithm continuously modifies the current solution. For each node, we utilize a binary variable indicating whether the node is selected as a facility and its traveling cost as features.

Moreover, the Voronoi cells associated with the facility set are natural structures formed in FLPs on a plane, providing inspiration for our approach. The 2D plane can be divided into $p$ regions corresponding to a set of $p$ facilities. Each region, termed a Voronoi cell, comprises points closest to the respective facility compared to others. Let $V_r$ denote the $r$-th Voronoi cell (associated with the $r$-th facility). We exploit additional information from the Voronoi cell created by each facility. Specifically, we compute two cell-based features for facility $f(r)$: the sum of demands in cell $\sum_{u \in P(V_r)} p_u$

**Algorithm 2:** Swap algorithm for $p$-median problem

---

**Parameters :** iteration number $T$, swap trials $K$, swap
budget $S$, swapping model *agent*
**Input:** graph $G$, facility number $p$
**Output:** facility set $F$

**1 Function** SwapLocate($T, K, S, p, agent$)
**2**    $F^* \leftarrow \emptyset$;
**3**    **for** $i \leftarrow 1$ **to** $T$ **do**
**4**      $F \leftarrow$ an initial set of $p$ facilities;
**5**      $(F_k, J_k), F \leftarrow$ SwapRelocate($K, agent, G, F, S$);
**6**      **if** $O(F) < O(F^*)$ **then**
**7**        $F^* \leftarrow F$;
**8**    **return** $F^*$;

---

and the total traveling cost in cell $\sum_{u \in P(V_r)} p_u d_{f(r)u}$. The facility features are padded with zero for non-facility nodes. Ablation experiments in Section 5.2 prove that Voronoi cells provide valuable insights into the quality of facility placement and guide the relocation process effectively. For node embedding, the concatenation of the aforementioned features serves as initial node features and is fed into GNN. The initial edge embeddings are the lengths of the edges. Let $\mathbf{h}_i^{L_1}$ denote the node embedding of node $i$ after the GNN module. We define a global embedding for the graph by $\mathbf{w}(G) = (\text{mean\_pooling}_{i=1}^V(\mathbf{h}_i^{L_1}) \| \text{max\_pooling}_{i=1}^V(\mathbf{h}_i^{L_1}))$, where $\|$ stands for vector concatenation. The critic MLP takes $\mathbf{w}(G)$ as input and yields a scalar to score the current state.

### 4.4 Attention-based Relocation Pair Selection

The action space for one relocation involves choosing two nodes: the facility to remove $u_1$ and the new facility to insert $u_2$, As the action space is quadratic to the number of nodes $(u_1, u_2) \in F \times (V \setminus F)$, we break down the target into a two-stage task, i.e. $P(u_1, u_2 | \Theta_{\mathcal{A}}) = P(u_1 | \Theta_{\mathcal{A}}) P(u_2 | \Theta_{\mathcal{A}}, u_1)$, where $\Theta_{\mathcal{A}}$ denotes the learnable parameters of the actor.

After the GNN module extracts an embedding $\mathbf{h}_i^{L_1}$ for each node $i$, a global embedding can be expressed by $\overline{\mathbf{h}} = \text{mean\_pooling}_{i=1}^V(\mathbf{h}_i^{L_1})$. Given the global context and node embeddings, subsequent $L_2$ layers of MLPs evaluate the priority of removing node (facility) $i$ by

$$\mathbf{g}_i^{L_2} = \text{MLP}(\mathbf{h}_i^{L_1} \| \overline{h}), \tag{7}$$

which yields the final probability distribution of removing $i$:

$$P^-(i | \Theta_{\mathcal{A}}) = \sigma(logit_1(i)), \quad logit_1(i) = \begin{cases} \mathbf{g}_i^{L_2}, & i \in F \\ -\infty, & i \notin F \end{cases}, \tag{8}$$

where $\sigma$ represents the Softmax function.

Let $u_1$ be the removed node sampled from $P^-$. Next, we consider choosing the new facility $u_2$ given node embeddings and $u_1$. This is implemented by an attention layer

$$\mathbf{f}_j = \text{Att}(\mathbf{h}_j^{L_1}) = (\mathbf{h}_j^{L_1})^\top \tanh(\text{Linear}(\mathbf{h}_{u_1}^{L_1})), \tag{9}$$

$$P^+(j | \Theta_{\mathcal{A}}, u_1) = \sigma(logit_2(j)), \quad logit_2(j) = \begin{cases} \mathbf{f}_j, & j \in V \setminus F \\ -\infty, & j \notin V \setminus F \end{cases}. \tag{10}$$

The inserted facility $u_2$ is sampled from $P^+$, completing the relocation pair $(u_1, u_2) \in F \times (V \setminus F)$.

### 4.5 From Relocation to Location

One advantage of Algorithm 1 is its versatility, allowing it to solve not only the facility relocation problem but also to extend seamlessly to the $p$-median problem. Algorithm 2 demonstrates how Algorithm 1 can be integrated as a subroutine within an interchange framework to address PMP. By introducing a hyper-parameter $S$ that controls the number of swaps, we can utilize the function SwapRelocate following the initial setup. The best solution obtained over $T$ trials is then returned as the final solution. Interestingly, employing a greedy agent for SwapRelocate results in the classical exchange algorithm [14], a well-studied heuristic for solving PMP [1].

## 5 Experiments

Previous works in the field often lack rigorous experimental setups, either not using separate test data [23] or evaluating models only on small instances [30], resulting in limited assessments of model performance. In contrast, our evaluation thoroughly assesses the efficiency and effectiveness of our algorithm for solving both FRP and PMP on complex graph data sets. Our model achieves a speedup of more than 2000 times to Gurobi on large instances while providing competitive solutions. Additional experiments show how our improving-style algorithm generalizes better than the constructive paradigm. Furthermore, the experiment on Shanghai road networks showcases how our method readily solves problems in real-world scenarios.

### 5.1 Solving Facility Relocation Problem on Weighted Graphs

Facility relocation is a useful modeling of problems that requires making limited modifications to an existing plan. For example, the bicycle-sharing system needs timely rebalancing to match users' traveling demands. To simulate the complex urban road networks, we construct a synthetic weighted graph data set based on Gabriel graphs [11], a type of planar graph that captures the geometric proximity of nodes. Node coordinates are generated with a bivariate normal distribution in $[0, 1]^2$. The demand for each node is generated randomly with the total demand controlled around 3,000,000.

*5.1.1 Baselines.* We implement two variants of Algorithm 1, namely Random-swap and Greedy-swap. Random-swap randomly selects relocation pairs and updates the solution if the new objective value is improved, representing the gain of relocation out of pure "luck". Greedy-swap always chooses the optimal swap at each step, selecting the pair that results in the greatest reduction in the objective function among all possible pairs. Furthermore, we compare two state-of-the-art heuristics for solving the facility relocation problem: the BestResponse algorithm from [21] based on Nash equilibrium and FR2FP from [31]. The BestResponse algorithm is adapted to align with our settings (see Appendix for details). The optimal solutions are computed by Gurobi [15] by setting MIPGap=0.

**Table 1: Results of Facility Relocation Problem**

| Methods | $n = 100$ | | | $n = 200$ | | | $n = 500$ | | | $n = 1000$ | | |
|---|---|---|---|---|---|---|---|---|---|---|---|---|
| | $Q$ (%) | Gap (%) | Time (s) | $Q$ (%) | Gap (%) | Time (s) | $Q$ (%) | Gap (%) | Time (s) | $Q$ (%) | Gap (%) | Time (s) |
| Gurobi | 51.77 | 0.00 | 0.15 | 51.90 | 0.00 | 0.94 | 63.39 | 0.00 | 12.15 | 70.09 | 0.00 | 128.20 |
| Greedy-swap | 51.26 | 1.09 | 1.67 | 51.37 | 1.15 | 6.50 | 62.76 | 1.83 | 30.08 | 69.60 | 1.80 | 99.52 |
| Random-swap | 25.22 | 72.86 | 0.04 | 25.54 | 68.46 | 0.04 | 31.49 | 100.63 | 0.07 | 36.80 | 125.76 | 0.08 |
| BestResponse | 49.21 | 6.27 | 1.49 | 49.36 | 6.37 | 4.60 | 61.32 | 6.04 | 21.89 | 68.89 | 4.22 | 68.77 |
| FR2FP | 46.53 | 10.87 | 0.11 | 46.18 | 13.22 | 0.24 | 59.54 | 10.77 | 0.86 | 66.78 | 11.21 | 3.42 |
| **PPO-swap** | 47.68 | 8.92 | 0.13 | 48.05 | 8.24 | 0.14 | 60.76 | 7.63 | 0.15 | 67.80 | 7.84 | 0.16 |

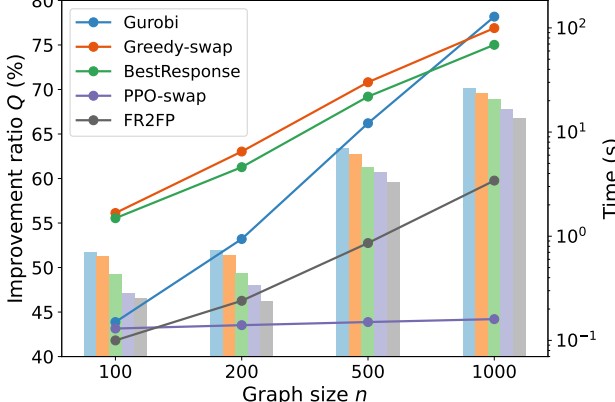

**Figure 3: FRP results on different scales.**

### 5.1.2 Settings.
To assess the efficacy and scalability of different methods, we conduct experiments with varying graph sizes, specifically $n \in [100, 200, 500, 1000]$. We generate 10 graphs of each graph size as our test set and evaluate the performance under various values of $p$, ranging from 5 to 40. For each instance, a random set of $p$ nodes is designated as the initial facility set, and the relocation budget is set to $\lfloor p/2 \rfloor$. Each undetermined algorithm runs for $T = 20$ iterations and records the best solution. PPO-swap is trained on 1000 graphs of size 100. The average results over $p$ are reported in Table 1, including improvement ratio $Q$ (defined in (5)), the optimality gap, and running time. The optimality gap of an improvement ratio $Q_1$ is defined as $\frac{Q_0 - Q_1}{Q_0} \times 100\%$, where $Q_0$ is the optimal improvement ratio. Higher $Q$-s and lower gaps are better.

### 5.1.3 Efficiency and Effectiveness Analysis.
Figure 3 visualizes the results in Table 1, with bars representing improvement ratios $Q$ and lines indicating the running times (in log scale). Greedy-swap is a strong heuristic and generates near-optimal solutions. However, it suffers rapid increases in computational overhead as the instance size and facility number grow, as it must iterate over an action space of size $p \times (n - p)$ at each step, which scales quadratically with the instance size. The BestResponse algorithm is also based on greedy strategies and has similar time complexity with slight speedup. The FR2FP algorithm falls in the middle with lower improvement ratios and higher efficiency.

PPO-swap stands out from peer methods with remarkable scalability and generalizability. Even if it was only trained on small instances, PPO-swap yields competitive solutions steadily for various graph sizes. Moreover, it performs fast inference and actions, taking under 0.2 seconds in all cases, with only slightly increasing running time for large instances. PPO-swap achieves a speedup of over 2000 compared to Gurobi when $n = 1000$, stressing its value in making real-time responses in times of emergency. The results of Random-swap, on the other hand, represent how much improvement comes from random swapping decisions, showcasing our model's ability to make wise choices.

## 5.2 Solving P-Median Problem on Weighted Graphs

This section evaluates the performance of PPO-swap on the $p$-median problem, where we choose $p$ facilities to minimize the global objective. We compare our method against established baselines on graph data sets. The evaluation focuses on solution quality and computational efficiency, highlighting the robustness of PPO-swap across different facility numbers.

### 5.2.1 Baselines.
As described in Section 4.5, three swap-based methods can be transplanted to solve PMP: Random-swap, Greedy-swap (essentially the interchange algorithm [14]), and PPO-swap. Additionally, we compare with a heuristic Maranzana [22] and a meta-heuristic simulated annealing (SA). We further introduce a variant of PPO-swap, namely PPO-no-vor, by replacing the Voronoi-based facility features introduced in Section 4.3 with zeros. This design of ablation is intended to prove the effectiveness of the features we have devised with domain knowledge. The optimal solution is computed by Gurobi [15] by setting MIPGap=0.

### 5.2.2 Settings.
The graph data sets replicate those in Section 5.1. Default hyper-parameters are set as follows: iteration number $T = 5$, swap trial $K = 20$, and swap budget $S = p$. SA runs for 1000 iterations. The average results over $p$ for different graph sizes are reported in Table 2, including the optimality gap and running time. The optimality gap of an objective $x$ and optimal objective $y$ is defined as $\frac{x - y}{y} \times 100\%$. Lower gaps are better.

### 5.2.3 Efficiency and Effectiveness Analysis.
Figure 4 illustrates five algorithms, with bars representing optimality gaps and lines indicating the running times (in log scale). Similarly, Gurobi and Greedy-swap produce high-quality solutions, but their running time grows exponentially with $n$, becoming intolerable for large

Table 2: Results of P-median Problem

| Methods | $n = 100$ | | $n = 200$ | | $n = 500$ | | $n = 1000$ | |
|---|---|---|---|---|---|---|---|---|
| | Gap (%) | Time (s) | Gap (%) | Time (s) | Gap (%) | Time (s) | Gap (%) | Time (s) |
| Gurobi | 0.00 | 0.14 | 0.00 | 0.88 | 0.00 | 13.43 | 0.00 | 126.40 |
| Greedy-swap | 0.07 | 10.04 | 0.09 | 48.69 | 0.10 | 234.92 | 0.11 | 733.11 |
| Random-swap | 25.94 | 0.33 | 27.41 | 0.39 | 27.62 | 0.60 | 26.97 | 0.84 |
| SA | 25.66 | 0.13 | 15.21 | 0.16 | 15.33 | 0.24 | 20.68 | 0.37 |
| Maranzana | 42.71 | 0.52 | 43.83 | 0.97 | 59.24 | 2.54 | 65.03 | 7.52 |
| **PPO-swap** | 6.36 | 1.33 | 8.35 | 1.38 | 8.87 | 1.40 | 10.16 | 1.53 |
| **PPO-no-vor** | 10.24 | 1.30 | 13.27 | 1.34 | 12.28 | 1.36 | 14.73 | 1.48 |

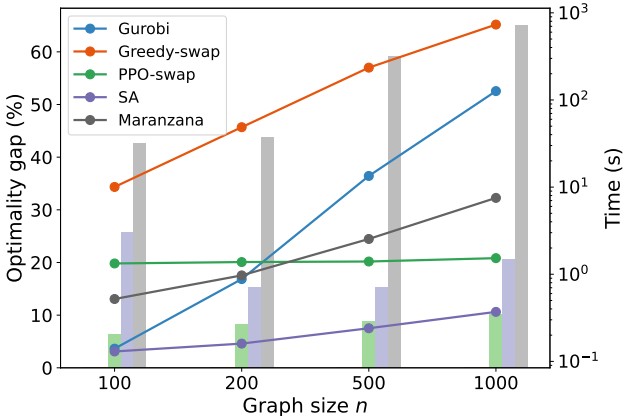

Figure 4: PMP results on different scales.

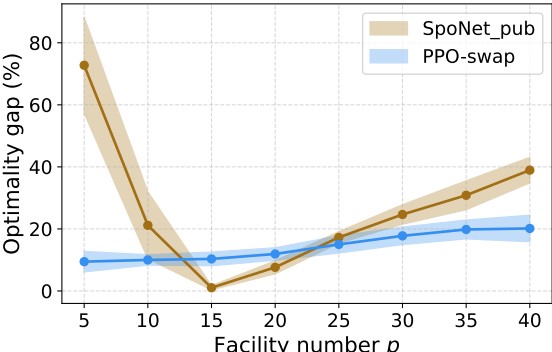

Figure 5: SpoNet_pub vs. PPO-swap on plane graphs.

graphs. PPO-swap, on the other hand, excels in generating stable and superior solutions even when faced with instances ten times larger than the training data. Its ability to maintain near-constant running times highlights its superior scalability compared to other heuristics, which often exhibit degraded performance and increased runtime as the problem size grows. The trade-off between solution quality and computational time favors PPO-swap in many cases, making it especially suitable for scenarios like emergency facility selection, where both rapid response and good solutions are crucial. By comparing PPO-no-vor and PPO-swap, we conclude that the Voronoi-based features indeed enhance both performance and stability, validating our model design.

## 5.3 Generalizability of Improving Algorithms

SpoNet is a latest model proposed by Liang et al. [18] that solves FLP with deep learning models in a constructive way. To compare the generalizability of two algorithm paradigms, we align with the PMP experiments in [18] and compare the optimality gaps of the two methods.

*5.3.1 Settings.* Each instance comprises 100 nodes with 2D coordinates uniformly distributed in the range $[0, 1]^2$. Nodes are connected if their distance is within a radius of 0.16. Given that [18] exclusively handles unweighted problems, we set the node demands to 1. The number of facilities is fixed at $p = 15$. PPO-swap is trained

on 1000 graphs following the same distribution, with 10 new graphs reserved for testing. We use the published model[1] from [18] for evaluation and denote it as SpoNet_pub. SpoNet_pub samples with a beam search width of 1280, and PPO-swap has the same hyperparameters as outlined in Section 5.2.

*5.3.2 Results and Analysis.* Figure 5 depicts the optimality gaps observed in SpoNet_pub and PPO-swap across the test data. Notably, while both models are trained with a fixed facility number $p$, SpoNet's constructive solving approach exhibits a greater dependency on this parameter. Consequently, it struggles to generalize effectively when this parameter changes. In contrast, PPO-swap demonstrates a smoother performance curve, indicating its superior adaptability to varying problem settings, a critical aspect of generalizability inherent in our improving-style algorithm.

## 5.4 Tackling Graph Complexity

In addressing the complexity of graph-based facility location problems, prior approaches [18, 30] rely on simplified graph constructions from 2D coordinates using Euclidean distances, while our method natively supports graph structures as direct input and incorporates non-Euclidean graph metrics. In these graphs, edges represent more complex metrics, such as shortest paths or travel times, reflecting real-world transportation networks with spatial

---

[1]https://github.com/CO-RL/SpoNet

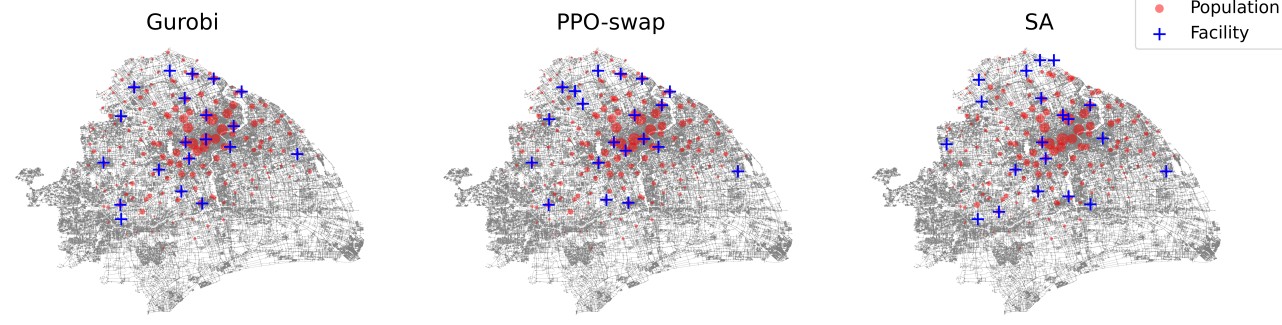

**Figure 6: PMP solutions of different methods on Shanghai road networks. The gray layer represents the urban road networks. The size of red circles is proportional to the regional population. Blue crosses stands for suggested locations for facilities.**

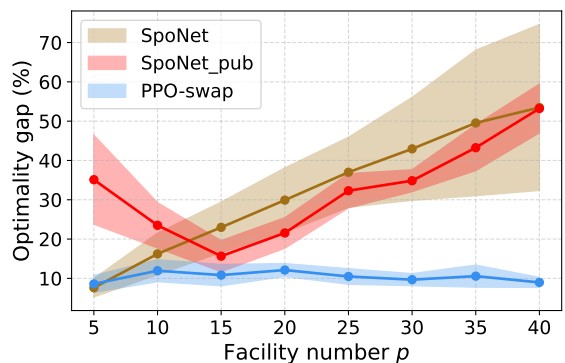

**Figure 7: SpoNet vs. PPO-swap on unweighted graphs.**

constraints. We devise the following experiments to demonstrate the non-triviality of this advance.

*5.4.1 Settings.* To demonstrate our method's effectiveness on complex graphs, we compared it against two variants of SpoNet [18]. The first one is SpoNet_pub as mentioned in Section 5.3, with its Euclidean distance matrix substituted with a pairwise shortest path matrix during inference. The second version SpoNet is trained by using the shortest path matrix as the cost matrix for various values of $p$ incrementally since it has to be trained with fixed $p$. We use unweighted graphs for training and testing since SpoNet does not support weighted graphs.

*5.4.2 Results and Analysis.* As illustrated in Figure 7, SpoNet achieves acceptable gaps when $p$ is small, but the performance declines as $p$ increases even with incremental training. For SpoNet_pub, its performance is consistent with the fact that it was trained with fixed $p = 15$. PPO-swap outperforms both versions of SpoNet significantly across different $p$. This experiment reveals the intrinsic advantage of our model, as it learns a general strategy to swap on complex graph data in spite of changes in instance parameters. This capability allows our method to capture richer graph-based information in applications, particularly when the objective is influenced by travel cost, a key factor in many road network applications.

## 5.5 Placing Facilities on Urban Road Networks

Besides the experiments on synthetic graphs, we demonstrate how PPO-swap can solve facility location problems in real-world scenarios. We use the road networks and population data of Shanghai to construct a city data set. Specifically, graph nodes consist of aggregated 5000m grids, and edges are connected based on the shape of the road network. For model training, we disturb the original population and generate 1000 sets of node weights. Other settings for inference are identical to Section 5.2.

Figure 6 illustrates the solutions of three methods on the city data set for $p = 20$, where the size of red circles denotes the amount of population and blue crosses are suggested locations to place facilities. Compared to SA, PPO-swap achieves lower costs and places facilities in a more efficient manner. As observed, PPO-swap increases the density of facilities in high-population areas, which reduces average travel distances for the population. Additionally, the spatial distribution of the PPO-swap solution is closer to that of the Gurobi solver, making it a practical and scalable solution for large-scale urban applications, especially for real-time deployment.

## 6 Conclusion

In conclusion, our work provides a novel and robust solution to facility location problems on graphs. We introduce a versatile swap-based framework addressing both the $p$-median problem and facility relocation on graphs, achieving a commendable balance between solution quality and running time. Extensive experiments on synthetic and real-world data sets show that it is capable of producing high-quality solutions on large graphs with fast inference.

Our work has certain limitations that warrant consideration for future research. First, there is room for improvement in performance, which could be achieved through the exploration of more intricate model architectures. Second, while our approach demonstrates competitive performance, the computation of node features may require acceleration to ensure that the model remains competitive with heuristics on small instances. Addressing these limitations could further enhance the applicability and efficiency of our approach in real-world settings.

In summary, our work contributes significantly to the field of facility location optimization, providing a robust framework that addresses complex challenges.

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

# A Implementation Details

## A.1 Implementation of PPO-swap

All experiments are conducted on an Intel Xeon Gold 6226R CPU with 64 cores and NVIDIA GeForce RTX 3090 GPU. The CPU operates on x86-64 architecture and the GPUs run CUDA version 12.1. For experiments on Gabriel graphs, we generate 1000 graphs with 100 nodes as training data set. We use the GNN implementation from Morris et al. [25]. It has $L_1 = 3$ layers with hidden size of 128. The edge embedding size is 1 and attention head is 1. For the MLP module, it has $L_2 = 3$ layers with hidden size of 128. We use PPO algorithm for reinforcement learning and Adam optimizer with learning rate of 0.005 and exponential decay rate of 0.995. The PPO training hyper-parameters are listed in Table 3.

**Table 3: Hyper-parameters for training PPO-swap**

| Hyper-parameters | Value |
|---|---|
| PPO gamma | 0.9 |
| GAE lambda | 0.95 |
| Batch size | 64 |
| Steps per epoch | 1024 |
| Optimization iterations | 4 |
| Clip ratio | 0.1 |
| Clip decay | 0.998 |
| Entropy loss weight | 0.01 |
| Critic loss weight | 0.5 |
| Gradient clipping | 1 |
| Number of epochs | 300 |

## A.2 Implementation of BestResponse

We use the BestResponse algorithm from [21] as a baseline for FRP with reasonable modification to align with our experiments. Since [21] optimizes a twofold objective of facility exposure and user convenience, it is slightly different to the setting of our work. We modify the BestResponse algorithm to optimize the global user-facility cost in Eqs. (3,4). We notice that BestResponse reaches comparable results with the RL method in [21] when considering single objective, and it should serve as a competitive baseline. Another difference in problem setting is that [21] does not limit the relocation budget, so we add this constraint to line 1.2 in the BestResponse algorithm. For a fair comparison, we choose relocated facilities randomly (in line 1.3) and run Best Response for 20 iterations (the same as other baselines).

## A.3 Implementation of SpoNet

The SpoNet [18] in Section 5.4 is modified to align with our settings. . First, we use the shortest path as the cost matrix in the model. Second, we trained SpoNet incrementally for varying values of the parameter $p$ (in range(5, 41, 5), each for 25 epochs, batch_size=64, and epoch_size=1024). These hyperparameters are aligned with PPO-swap.

# B Additional Experiment Results

## B.1 Grid Search results for GNN Architectures

We perform grid search over three hyperparameters concerning the GNN architectures: the types of GNN layer, the number of hidden size and the number of GNN layers. All combinations are trained while other hyperparameters are fixed. Table 4 reports the improvement ratio $Q$ of each model on the test set for facility relocation problem, where the combination (`GraphConv, 128, 3`) performs best.

## B.2 Experiments for FRP on Various Scales

The detailed results for $n = 100, 200, 500, 1000$ solving FRP are shown in Table 5,6,7,8. The model used for PPO-swap is trained on graphs with $n = 100$ only.

## B.3 Experiments for PMP on Various Scales

The detailed results for $n = 100, 200, 500, 1000$ solving PMP are shown in Table 9,10,11,12. The model used for PPO-swap is trained on graphs with $n = 100$ only.

Received 20 February 2007; revised 12 March 2009; accepted 5 June 2009

**Table 4: Grid search of GNN architecture hyperparamters**

| layer_name | (32, 2) | (32, 3) | (32, 4) | (64, 2) | (64, 3) | (64, 4) | (128, 2) | (128, 3) | (128, 4) |
|---|---|---|---|---|---|---|---|---|---|
| GATv2Conv [3] | 46.58 | 46.30 | 44.01 | 45.57 | 46.10 | 45.13 | 46.81 | 45.73 | 44.60 |
| GCNConv [17] | 45.73 | 44.58 | 42.88 | 45.84 | 44.25 | 43.20 | 45.01 | 43.87 | 43.24 |
| GraphConv [25] | 46.85 | 46.77 | 47.01 | 46.17 | 47.41 | 47.10 | 42.03 | **47.44** | 47.18 |

**Table 5: Results of FRP, $n = 100$**

| Methods | $p = 5$ | | | $p = 10$ | | | $p = 15$ | | | $p = 20$ | | |
|---|---|---|---|---|---|---|---|---|---|---|---|---|
| | Q (%) | Gap (%) | Time (s) | Q (%) | Gap (%) | Time (s) | Q (%) | Gap (%) | Time (s) | Q (%) | Gap (%) | Time (s) |
| Gurobi | 39.70 ± 11.19 | 0.00 ± 0.00 | 0.33 ± 0.04 | 42.16 ± 9.44 | 0.00 ± 0.00 | 0.36 ± 0.17 | 47.36 ± 9.11 | 0.00 ± 0.00 | 0.36 ± 0.14 | 47.33 ± 9.66 | 0.00 ± 0.00 | 0.47 ± 0.30 |
| Greedy-swap | 39.00 ± 10.70 | 1.44 ± 2.80 | 0.11 ± 0.03 | 41.23 ± 9.67 | 1.57 ± 1.39 | 0.41 ± 0.09 | 47.12 ± 9.12 | 0.48 ± 0.42 | 0.91 ± 0.25 | 46.91 ± 9.67 | 0.81 ± 0.89 | 1.69 ± 0.29 |
| Random-swap | 29.92 ± 12.39 | 17.04 ± 11.85 | 0.01 ± 0.00 | 22.72 ± 8.14 | 36.33 ± 21.99 | 0.03 ± 0.01 | 27.44 ± 7.12 | 40.59 ± 19.41 | 0.04 ± 0.01 | 21.54 ± 8.47 | 52.11 ± 22.77 | 0.05 ± 0.02 |
| BestResponse | 38.92 ± 10.51 | 1.63 ± 2.62 | 0.28 ± 0.02 | 40.29 ± 9.84 | 3.23 ± 2.13 | 0.59 ± 0.06 | 44.94 ± 8.97 | 4.79 ± 1.47 | 0.79 ± 0.07 | 44.86 ± 10.05 | 4.71 ± 2.01 | 1.15 ± 0.12 |
| FR2FP | 33.93 ± 12.59 | 9.69 ± 6.28 | 0.02 ± 0.00 | 34.86 ± 11.34 | 12.61 ± 5.96 | 0.07 ± 0.01 | 39.56 ± 8.95 | 15.42 ± 6.09 | 0.10 ± 0.01 | 41.73 ± 11.51 | 10.44 ± 3.94 | 0.13 ± 0.02 |
| **PPO-swap** | 38.38 ± 11.35 | 2.21 ± 1.80 | 0.03 ± 0.00 | 37.68 ± 11.60 | 7.32 ± 3.52 | 0.06 ± 0.00 | 42.62 ± 10.35 | 8.89 ± 2.32 | 0.09 ± 0.00 | 42.53 ± 11.17 | 8.90 ± 3.95 | 0.12 ± 0.00 |

| Methods | $p = 25$ | | | $p = 30$ | | | $p = 35$ | | | $p = 40$ | | |
|---|---|---|---|---|---|---|---|---|---|---|---|---|
| | Q (%) | Gap (%) | Time (s) | Q (%) | Gap (%) | Time (s) | Q (%) | Gap (%) | Time (s) | Q (%) | Gap (%) | Time (s) |
| Gurobi | 54.65 ± 13.45 | 0.00 ± 0.00 | 0.34 ± 0.09 | 54.91 ± 12.81 | 0.00 ± 0.00 | 0.34 ± 0.09 | 61.67 ± 12.70 | 0.00 ± 0.00 | 0.30 ± 0.09 | 66.36 ± 11.38 | 0.00 ± 0.00 | 0.26 ± 0.03 |
| Greedy-swap | 54.07 ± 13.64 | 1.24 ± 0.84 | 2.58 ± 0.54 | 54.41 ± 12.83 | 1.30 ± 1.00 | 3.76 ± 0.87 | 61.30 ± 12.73 | 1.06 ± 0.79 | 9.86 ± 2.28 | 66.07 ± 11.48 | 0.85 ± 0.67 | 12.60 ± 3.33 |
| Random-swap | 25.96 ± 8.72 | 77.02 ± 51.62 | 0.06 ± 0.01 | 23.81 ± 10.06 | 86.46 ± 69.85 | 0.07 ± 0.01 | 27.04 ± 9.07 | 116.43 ± 94.27 | 0.14 ± 0.06 | 26.88 ± 8.80 | 156.17 ± 126.53 | 0.16 ± 0.07 |
| BestResponse | 51.64 ± 13.67 | 7.23 ± 2.49 | 1.36 ± 0.21 | 51.39 ± 13.27 | 8.45 ± 3.17 | 1.76 ± 0.18 | 57.72 ± 12.87 | 11.65 ± 5.24 | 3.14 ± 0.27 | 63.57 ± 11.90 | 9.15 ± 2.93 | 3.71 ± 0.36 |
| FR2FP | 49.75 ± 14.97 | 10.53 ± 4.05 | 0.19 ± 0.04 | 51.21 ± 13.75 | 8.51 ± 3.06 | 0.25 ± 0.04 | 57.93 ± 13.56 | 10.05 ± 3.08 | 0.31 ± 0.05 | 63.22 ± 12.25 | 9.68 ± 4.17 | 0.47 ± 0.06 |
| **PPO-swap** | 50.01 ± 16.00 | 9.62 ± 4.17 | 0.14 ± 0.00 | 48.80 ± 15.46 | 12.98 ± 4.26 | 0.18 ± 0.00 | 56.68 ± 15.00 | 12.38 ± 3.93 | 0.20 ± 0.00 | 60.65 ± 13.50 | 16.79 ± 2.44 | 0.23 ± 0.00 |

**Table 6: Results of FRP, $n = 200$**

| Methods | $p = 5$ | | | $p = 10$ | | | $p = 15$ | | | $p = 20$ | | |
|---|---|---|---|---|---|---|---|---|---|---|---|---|
| | Q (%) | Gap (%) | Time (s) | Q (%) | Gap (%) | Time (s) | Q (%) | Gap (%) | Time (s) | Q (%) | Gap (%) | Time (s) |
| Gurobi | 39.70 ± 11.19 | 0.00 ± 0.00 | 0.15 ± 0.01 | 42.16 ± 9.44 | 0.00 ± 0.00 | 0.16 ± 0.07 | 47.36 ± 9.11 | 0.00 ± 0.00 | 0.15 ± 0.05 | 47.33 ± 9.66 | 0.00 ± 0.00 | 0.19 ± 0.11 |
| Greedy-swap | 39.00 ± 10.70 | 1.44 ± 2.80 | 0.06 ± 0.00 | 41.23 ± 9.67 | 1.57 ± 1.39 | 0.27 ± 0.01 | 47.12 ± 9.12 | 0.48 ± 0.42 | 0.52 ± 0.01 | 46.91 ± 9.67 | 0.81 ± 0.89 | 0.97 ± 0.01 |
| Random-swap | 26.63 ± 12.25 | 22.42 ± 8.50 | 0.01 ± 0.00 | 24.11 ± 8.95 | 32.80 ± 13.82 | 0.02 ± 0.00 | 25.91 ± 6.07 | 43.42 ± 17.40 | 0.02 ± 0.00 | 21.46 ± 7.13 | 52.46 ± 21.96 | 0.03 ± 0.00 |
| BestResponse | 38.72 ± 10.48 | 1.97 ± 2.58 | 0.28 ± 0.01 | 40.50 ± 9.66 | 2.88 ± 1.65 | 0.55 ± 0.02 | 45.72 ± 9.00 | 3.28 ± 1.63 | 0.75 ± 0.03 | 44.67 ± 9.41 | 5.30 ± 2.35 | 1.10 ± 0.07 |
| FR2FP | 33.93 ± 12.59 | 9.69 ± 6.28 | 0.01 ± 0.00 | 34.86 ± 11.34 | 12.61 ± 5.96 | 0.04 ± 0.01 | 39.56 ± 8.95 | 15.42 ± 6.09 | 0.07 ± 0.01 | 41.73 ± 11.51 | 10.44 ± 3.94 | 0.09 ± 0.01 |
| **PPO-swap** | 38.89 ± 11.09 | 1.44 ± 1.24 | 0.03 ± 0.00 | 38.54 ± 11.91 | 5.73 ± 3.49 | 0.06 ± 0.00 | 43.38 ± 10.28 | 7.45 ± 2.34 | 0.09 ± 0.00 | 43.06 ± 10.26 | 8.17 ± 2.27 | 0.12 ± 0.00 |

| Methods | $p = 25$ | | | $p = 30$ | | | $p = 35$ | | | $p = 40$ | | |
|---|---|---|---|---|---|---|---|---|---|---|---|---|
| | Q (%) | Gap (%) | Time (s) | Q (%) | Gap (%) | Time (s) | Q (%) | Gap (%) | Time (s) | Q (%) | Gap (%) | Time (s) |
| Gurobi | 54.65 ± 13.45 | 0.00 ± 0.00 | 0.14 ± 0.04 | 54.91 ± 12.81 | 0.00 ± 0.00 | 0.16 ± 0.04 | 61.67 ± 12.70 | 0.00 ± 0.00 | 0.13 ± 0.04 | 66.36 ± 11.38 | 0.00 ± 0.00 | 0.11 ± 0.01 |
| Greedy-swap | 54.07 ± 13.64 | 1.24 ± 0.84 | 1.42 ± 0.02 | 54.41 ± 12.83 | 1.30 ± 1.00 | 2.07 ± 0.02 | 61.30 ± 12.73 | 1.06 ± 0.79 | 3.54 ± 0.03 | 66.07 ± 11.48 | 0.85 ± 0.67 | 4.53 ± 0.02 |
| Random-swap | 25.83 ± 7.43 | 80.08 ± 64.70 | 0.04 ± 0.00 | 23.12 ± 12.24 | 83.51 ± 51.26 | 0.05 ± 0.00 | 24.77 ± 7.41 | 123.80 ± 99.09 | 0.06 ± 0.00 | 29.93 ± 9.71 | 144.37 ± 121.66 | 0.07 ± 0.00 |
| BestResponse | 51.28 ± 13.31 | 8.31 ± 3.57 | 1.29 ± 0.10 | 51.95 ± 12.95 | 7.33 ± 3.57 | 1.66 ± 0.09 | 58.10 ± 12.98 | 10.08 ± 3.99 | 3.01 ± 0.14 | 62.77 ± 12.26 | 10.97 ± 3.69 | 3.27 ± 0.25 |
| FR2FP | 49.75 ± 14.97 | 10.53 ± 4.05 | 0.09 ± 0.01 | 51.21 ± 13.75 | 8.51 ± 3.06 | 0.12 ± 0.01 | 57.93 ± 13.56 | 10.05 ± 3.08 | 0.15 ± 0.01 | 63.22 ± 12.25 | 9.68 ± 4.17 | 0.27 ± 0.01 |
| **PPO-swap** | 49.84 ± 15.53 | 10.34 ± 3.15 | 0.14 ± 0.00 | 49.09 ± 15.15 | 12.34 ± 4.17 | 0.18 ± 0.00 | 57.07 ± 14.42 | 11.66 ± 2.83 | 0.20 ± 0.00 | 61.58 ± 12.98 | 14.25 ± 2.86 | 0.23 ± 0.00 |

**Table 7: Results of FRP, $n = 500$**

| Methods | $p = 5$ | | | $p = 10$ | | | $p = 15$ | | | $p = 20$ | | |
|---|---|---|---|---|---|---|---|---|---|---|---|---|
| | Q (%) | Gap (%) | Time (s) | Q (%) | Gap (%) | Time (s) | Q (%) | Gap (%) | Time (s) | Q (%) | Gap (%) | Time (s) |
| Gurobi | 48.50 ± 13.98 | 0.00 ± 0.00 | 11.99 ± 2.66 | 64.63 ± 12.62 | 0.00 ± 0.00 | 14.28 ± 3.81 | 56.32 ± 7.58 | 0.00 ± 0.00 | 13.14 ± 4.16 | 62.43 ± 8.36 | 0.00 ± 0.00 | 12.08 ± 3.38 |
| Greedy-swap | 47.88 ± 13.86 | 1.35 ± 2.28 | 0.61 ± 0.02 | 64.04 ± 12.58 | 1.92 ± 1.39 | 3.91 ± 0.04 | 55.48 ± 7.76 | 1.96 ± 1.38 | 8.86 ± 0.54 | 62.00 ± 8.50 | 1.11 ± 0.90 | 18.01 ± 1.52 |
| Random-swap | 30.37 ± 16.18 | 37.61 ± 19.98 | 0.01 ± 0.00 | 43.27 ± 16.34 | 64.14 ± 19.09 | 0.03 ± 0.01 | 27.66 ± 12.67 | 66.15 ± 16.40 | 0.04 ± 0.01 | 27.30 ± 9.32 | 99.90 ± 32.44 | 0.06 ± 0.01 |
| BestResponse | 47.88 ± 13.85 | 1.34 ± 2.28 | 2.36 ± 0.06 | 63.51 ± 12.94 | 3.24 ± 1.34 | 7.90 ± 0.21 | 53.95 ± 8.18 | 5.41 ± 1.76 | 11.77 ± 0.16 | 59.98 ± 9.22 | 6.28 ± 2.35 | 18.09 ± 1.16 |
| FR2FP | 43.77 ± 17.26 | 8.55 ± 9.10 | 0.10 ± 0.02 | 62.18 ± 13.18 | 7.24 ± 3.29 | 0.28 ± 0.02 | 52.43 ± 8.62 | 8.84 ± 2.91 | 0.44 ± 0.03 | 58.57 ± 9.13 | 10.50 ± 3.55 | 0.69 ± 0.03 |
| **PPO-swap** | 47.38 ± 14.85 | 1.90 ± 1.49 | 0.03 ± 0.00 | 62.41 ± 12.89 | 6.93 ± 4.18 | 0.07 ± 0.01 | 53.82 ± 7.87 | 5.83 ± 2.12 | 0.10 ± 0.01 | 59.37 ± 9.08 | 8.11 ± 2.61 | 0.14 ± 0.01 |

| Methods | $p = 25$ | | | $p = 30$ | | | $p = 35$ | | | $p = 40$ | | |
|---|---|---|---|---|---|---|---|---|---|---|---|---|
| | Q (%) | Gap (%) | Time (s) | Q (%) | Gap (%) | Time (s) | Q (%) | Gap (%) | Time (s) | Q (%) | Gap (%) | Time (s) |
| Gurobi | 69.79 ± 9.18 | 0.00 ± 0.00 | 10.73 ± 1.77 | 67.09 ± 10.91 | 0.00 ± 0.00 | 10.90 ± 1.62 | 67.69 ± 10.55 | 0.00 ± 0.00 | 11.73 ± 1.66 | 70.63 ± 7.22 | 0.00 ± 0.00 | 12.35 ± 2.67 |
| Greedy-swap | 69.14 ± 9.16 | 2.34 ± 0.99 | 27.33 ± 0.35 | 66.43 ± 11.04 | 2.25 ± 1.43 | 43.43 ± 0.77 | 67.02 ± 10.79 | 2.06 ± 0.67 | 58.20 ± 1.34 | 70.12 ± 7.35 | 1.69 ± 1.19 | 80.30 ± 1.27 |
| Random-swap | 33.80 ± 13.32 | 127.94 ± 37.46 | 0.08 ± 0.01 | 34.18 ± 16.16 | 110.38 ± 36.87 | 0.10 ± 0.02 | 27.39 ± 12.62 | 140.11 ± 49.90 | 0.11 ± 0.02 | 27.95 ± 6.91 | 158.78 ± 57.72 | 0.13 ± 0.02 |
| BestResponse | 67.87 ± 9.75 | 6.32 ± 2.39 | 23.09 ± 1.23 | 64.76 ± 11.42 | 7.42 ± 2.83 | 31.47 ± 3.52 | 64.65 ± 11.68 | 9.34 ± 2.51 | 37.53 ± 7.21 | 67.96 ± 7.96 | 8.94 ± 2.28 | 42.89 ± 3.46 |
| FR2FP | 66.24 ± 10.07 | 11.95 ± 2.86 | 0.89 ± 0.10 | 63.27 ± 11.89 | 11.90 ± 3.34 | 1.17 ± 0.04 | 63.36 ± 12.08 | 13.36 ± 5.11 | 1.45 ± 0.09 | 66.50 ± 8.52 | 13.82 ± 3.47 | 1.85 ± 0.14 |
| **PPO-swap** | 67.08 ± 10.05 | 8.89 ± 1.43 | 0.17 ± 0.01 | 63.50 ± 12.51 | 10.47 ± 3.17 | 0.21 ± 0.01 | 64.68 ± 11.96 | 8.98 ± 2.19 | 0.23 ± 0.01 | 67.85 ± 7.65 | 9.95 ± 5.17 | 0.27 ± 0.01 |

## Table 8: Results of FRP, $n = 1000$

| Methods | $p = 5$ | | | $p = 10$ | | | $p = 15$ | | | $p = 20$ | | |
|---|---|---|---|---|---|---|---|---|---|---|---|---|
| | $Q$ (%) | Gap (%) | Time (s) | $Q$ (%) | Gap (%) | Time (s) | $Q$ (%) | Gap (%) | Time (s) | $Q$ (%) | Gap (%) | Time (s) |
| Gurobi | 59.87 ± 13.91 | 0.00 ± 0.00 | 111.32 ± 45.98 | 70.22 ± 6.15 | 0.00 ± 0.00 | 151.85 ± 50.54 | 65.89 ± 10.45 | 0.00 ± 0.00 | 141.85 ± 51.53 | 70.11 ± 8.34 | 0.00 ± 0.00 | 130.52 ± 33.89 |
| Greedy-swap | 59.70 ± 13.92 | 0.46 ± 0.65 | 3.56 ± 0.62 | 69.42 ± 6.22 | 2.75 ± 1.18 | 19.23 ± 3.32 | 65.63 ± 10.31 | 1.02 ± 0.96 | 42.08 ± 8.54 | 69.52 ± 8.37 | 2.17 ± 1.39 | 87.70 ± 14.86 |
| Random-swap | 34.74 ± 18.20 | 69.63 ± 36.49 | 0.02 ± 0.00 | 41.11 ± 9.27 | 103.30 ± 40.54 | 0.04 ± 0.00 | 39.24 ± 16.04 | 82.27 ± 20.70 | 0.06 ± 0.00 | 37.07 ± 14.90 | 115.85 ± 26.77 | 0.08 ± 0.01 |
| BestResponse | 59.70 ± 13.92 | 0.46 ± 0.65 | 11.87 ± 3.50 | 69.38 ± 6.26 | 2.88 ± 1.15 | 33.09 ± 8.01 | 65.05 ± 10.62 | 2.60 ± 0.90 | 50.37 ± 11.15 | 68.94 ± 8.57 | 4.09 ± 1.29 | 77.60 ± 18.87 |
| FR2FP | 56.35 ± 16.37 | 7.52 ± 5.96 | 0.30 ± 0.08 | 68.80 ± 6.19 | 4.94 ± 3.56 | 1.16 ± 0.23 | 62.95 ± 11.04 | 8.71 ± 2.89 | 1.85 ± 0.44 | 66.83 ± 9.46 | 10.89 ± 4.33 | 2.65 ± 0.68 |
| **PPO-swap** | 58.84 ± 14.26 | 2.54 ± 3.28 | 0.03 ± 0.00 | 68.54 ± 6.61 | 5.51 ± 1.77 | 0.07 ± 0.00 | 63.85 ± 10.84 | 6.26 ± 2.13 | 0.10 ± 0.00 | 67.15 ± 9.64 | 9.40 ± 2.98 | 0.14 ± 0.00 |

| Methods | $p = 25$ | | | $p = 30$ | | | $p = 35$ | | | $p = 40$ | | |
|---|---|---|---|---|---|---|---|---|---|---|---|---|
| | $Q$ (%) | Gap (%) | Time (s) | $Q$ (%) | Gap (%) | Time (s) | $Q$ (%) | Gap (%) | Time (s) | $Q$ (%) | Gap (%) | Time (s) |
| Gurobi | 72.62 ± 8.58 | 0.00 ± 0.00 | 109.81 ± 24.68 | 75.36 ± 7.77 | 0.00 ± 0.00 | 122.62 ± 25.28 | 70.10 ± 9.14 | 0.00 ± 0.00 | 124.17 ± 34.47 | 76.59 ± 8.40 | 0.00 ± 0.00 | 133.45 ± 39.36 |
| Greedy-swap | 71.93 ± 8.77 | 2.56 ± 1.44 | 134.23 ± 18.64 | 74.90 ± 8.04 | 1.73 ± 0.67 | 206.06 ± 32.79 | 69.48 ± 9.40 | 2.01 ± 1.17 | 129.97 ± 15.11 | 76.20 ± 8.53 | 1.72 ± 0.97 | 173.33 ± 26.67 |
| Random-swap | 38.06 ± 12.30 | 135.29 ± 35.62 | 0.10 ± 0.01 | 39.43 ± 12.47 | 154.94 ± 36.73 | 0.13 ± 0.01 | 27.79 ± 12.09 | 155.49 ± 50.42 | 0.10 ± 0.00 | 36.96 ± 13.11 | 189.31 ± 61.93 | 0.12 ± 0.00 |
| BestResponse | 71.03 ± 9.26 | 5.59 ± 1.56 | 93.43 ± 17.85 | 74.13 ± 8.49 | 4.68 ± 1.81 | 132.85 ± 23.07 | 67.81 ± 10.21 | 7.23 ± 3.14 | 68.51 ± 5.31 | 75.13 ± 8.98 | 6.21 ± 1.81 | 82.45 ± 7.32 |
| FR2FP | 69.18 ± 9.95 | 12.20 ± 2.85 | 3.72 ± 0.89 | 71.84 ± 9.40 | 13.79 ± 5.49 | 5.16 ± 1.34 | 65.13 ± 10.54 | 16.89 ± 3.54 | 5.70 ± 1.49 | 73.16 ± 9.51 | 14.73 ± 2.76 | 6.83 ± 1.42 |
| **PPO-swap** | 70.06 ± 9.83 | 8.92 ± 2.61 | 0.17 ± 0.00 | 72.79 ± 8.91 | 10.07 ± 3.89 | 0.22 ± 0.00 | 67.18 ± 10.53 | 9.32 ± 3.09 | 0.24 ± 0.00 | 73.98 ± 9.49 | 10.74 ± 1.99 | 0.28 ± 0.02 |

## Table 9: Results of PMP, $n = 100$

| Methods | $p = 5$ | | $p = 10$ | | $p = 15$ | | $p = 20$ | |
|---|---|---|---|---|---|---|---|---|
| | Gap (%) | Time (s) | Gap (%) | Time (s) | Gap (%) | Time (s) | Gap (%) | Time (s) |
| Gurobi | 0.00 ± 0.00 | 0.16 ± 0.05 | 0.00 ± 0.00 | 0.15 ± 0.03 | 0.00 ± 0.00 | 0.15 ± 0.04 | 0.00 ± 0.00 | 0.16 ± 0.07 |
| Greedy-swap | 0.00 ± 0.00 | 0.71 ± 0.04 | 0.02 ± 0.06 | 2.32 ± 0.22 | 0.01 ± 0.03 | 4.87 ± 0.68 | 0.18 ± 0.28 | 7.19 ± 1.19 |
| Random-swap | 10.92 ± 3.78 | 0.08 ± 0.00 | 14.66 ± 3.25 | 0.14 ± 0.01 | 18.06 ± 4.30 | 0.16 ± 0.00 | 22.01 ± 3.07 | 0.22 ± 0.00 |
| SA | 8.14 ± 3.81 | 0.15 ± 0.01 | 8.62 ± 3.15 | 0.11 ± 0.00 | 11.72 ± 3.36 | 0.10 ± 0.00 | 15.00 ± 6.20 | 0.10 ± 0.00 |
| Maranzana | 5.68 ± 3.24 | 0.26 ± 0.01 | 15.55 ± 12.05 | 0.29 ± 0.03 | 21.65 ± 17.27 | 0.36 ± 0.07 | 30.53 ± 17.63 | 0.44 ± 0.09 |
| **PPO-swap** | 3.81 ± 2.01 | 0.31 ± 0.01 | 7.24 ± 2.56 | 0.60 ± 0.01 | 6.18 ± 1.93 | 0.90 ± 0.01 | 5.64 ± 2.21 | 1.19 ± 0.02 |
| **PPO-no-vor** | 5.34 ± 2.43 | 0.31 ± 0.01 | 8.47 ± 2.66 | 0.59 ± 0.01 | 10.46 ± 2.06 | 0.88 ± 0.01 | 10.76 ± 2.80 | 1.16 ± 0.02 |

| Methods | $p = 25$ | | $p = 30$ | | $p = 35$ | | $p = 40$ | |
|---|---|---|---|---|---|---|---|---|
| | Gap (%) | Time (s) | Gap (%) | Time (s) | Gap (%) | Time (s) | Gap (%) | Time (s) |
| Gurobi | 0.00 ± 0.00 | 0.12 ± 0.01 | 0.00 ± 0.00 | 0.14 ± 0.04 | 0.00 ± 0.00 | 0.13 ± 0.04 | 0.00 ± 0.00 | 0.10 ± 0.00 |
| Greedy-swap | 0.14 ± 0.29 | 10.29 ± 1.93 | 0.07 ± 0.11 | 12.09 ± 2.53 | 0.06 ± 0.18 | 20.56 ± 4.69 | 0.08 ± 0.18 | 22.30 ± 4.93 |
| Random-swap | 30.87 ± 4.92 | 0.28 ± 0.00 | 32.95 ± 5.02 | 0.35 ± 0.01 | 36.15 ± 7.07 | 0.68 ± 0.01 | 41.90 ± 7.92 | 0.75 ± 0.01 |
| SA | 19.88 ± 4.99 | 0.11 ± 0.00 | 42.40 ± 53.01 | 0.11 ± 0.00 | 41.75 ± 32.80 | 0.20 ± 0.00 | 57.74 ± 33.02 | 0.20 ± 0.01 |
| Maranzana | 44.54 ± 34.59 | 0.49 ± 0.08 | 55.04 ± 40.54 | 0.54 ± 0.13 | 73.15 ± 60.72 | 0.73 ± 0.10 | 95.54 ± 90.67 | 1.05 ± 0.43 |
| **PPO-swap** | 7.34 ± 1.38 | 1.48 ± 0.02 | 6.94 ± 2.02 | 1.78 ± 0.03 | 7.45 ± 1.31 | 2.04 ± 0.03 | 6.32 ± 1.39 | 2.34 ± 0.03 |
| **PPO-no-vor** | 12.05 ± 2.41 | 1.45 ± 0.02 | 11.35 ± 2.77 | 1.74 ± 0.02 | 11.08 ± 3.35 | 1.98 ± 0.03 | 12.40 ± 4.48 | 2.27 ± 0.03 |

**Table 10: Results of PMP, $n = 200$**

| Methods | $p = 5$ | | $p = 10$ | | $p = 15$ | | $p = 20$ | |
|---|---|---|---|---|---|---|---|---|
| | Gap (%) | Time (s) | Gap (%) | Time (s) | Gap (%) | Time (s) | Gap (%) | Time (s) |
| Gurobi | 0.00 ± 0.00 | 1.12 ± 0.49 | 0.00 ± 0.00 | 0.87 ± 0.20 | 0.00 ± 0.00 | 1.00 ± 0.38 | 0.00 ± 0.00 | 0.86 ± 0.23 |
| Greedy-swap | 0.13 ± 0.21 | 1.79 ± 0.14 | 0.15 ± 0.25 | 6.93 ± 0.21 | 0.03 ± 0.06 | 14.72 ± 1.27 | 0.04 ± 0.09 | 39.85 ± 10.06 |
| Random-swap | 9.00 ± 4.99 | 0.08 ± 0.00 | 20.45 ± 2.88 | 0.15 ± 0.00 | 25.20 ± 3.93 | 0.23 ± 0.00 | 26.95 ± 5.94 | 0.34 ± 0.00 |
| SA | 4.69 ± 1.78 | 0.13 ± 0.00 | 8.47 ± 3.58 | 0.14 ± 0.00 | 11.01 ± 4.08 | 0.14 ± 0.00 | 13.45 ± 5.29 | 0.16 ± 0.01 |
| Maranzana | 7.42 ± 4.50 | 0.37 ± 0.02 | 14.81 ± 8.35 | 0.59 ± 0.08 | 27.47 ± 17.96 | 0.72 ± 0.11 | 37.67 ± 21.65 | 0.97 ± 0.17 |
| **PPO-swap** | 4.48 ± 1.46 | 0.33 ± 0.01 | 6.97 ± 2.19 | 0.63 ± 0.01 | 8.69 ± 2.68 | 0.93 ± 0.01 | 9.21 ± 2.26 | 1.23 ± 0.02 |
| **PPO-no-vor** | 6.56 ± 5.07 | 0.33 ± 0.01 | 11.69 ± 4.52 | 0.62 ± 0.01 | 13.29 ± 2.38 | 0.91 ± 0.02 | 13.91 ± 2.40 | 1.21 ± 0.02 |

| Methods | $p = 25$ | | $p = 30$ | | $p = 35$ | | $p = 40$ | |
|---|---|---|---|---|---|---|---|---|
| | Gap (%) | Time (s) | Gap (%) | Time (s) | Gap (%) | Time (s) | Gap (%) | Time (s) |
| Gurobi | 0.00 ± 0.00 | 0.79 ± 0.29 | 0.00 ± 0.00 | 0.76 ± 0.22 | 0.00 ± 0.00 | 0.84 ± 0.36 | 0.00 ± 0.00 | 0.81 ± 0.33 |
| Greedy-swap | 0.12 ± 0.18 | 53.52 ± 13.38 | 0.07 ± 0.11 | 74.56 ± 19.60 | 0.03 ± 0.06 | 93.46 ± 20.31 | 0.14 ± 0.17 | 104.71 ± 30.22 |
| Random-swap | 30.57 ± 4.01 | 0.44 ± 0.00 | 31.89 ± 3.82 | 0.54 ± 0.00 | 36.33 ± 7.88 | 0.61 ± 0.00 | 38.87 ± 8.94 | 0.72 ± 0.00 |
| SA | 16.67 ± 6.98 | 0.16 ± 0.00 | 17.76 ± 6.74 | 0.17 ± 0.00 | 21.38 ± 9.29 | 0.17 ± 0.00 | 28.23 ± 20.58 | 0.17 ± 0.00 |
| Maranzana | 48.25 ± 39.37 | 1.07 ± 0.24 | 62.09 ± 52.38 | 1.33 ± 0.27 | 72.72 ± 58.88 | 1.36 ± 0.24 | 80.21 ± 63.38 | 1.36 ± 0.28 |
| **PPO-swap** | 8.63 ± 2.97 | 1.54 ± 0.02 | 9.59 ± 2.33 | 1.86 ± 0.03 | 9.30 ± 1.68 | 2.11 ± 0.03 | 9.94 ± 2.22 | 2.42 ± 0.03 |
| **PPO-no-vor** | 15.19 ± 2.68 | 1.50 ± 0.02 | 14.35 ± 3.22 | 1.80 ± 0.04 | 15.31 ± 2.80 | 2.05 ± 0.03 | 15.87 ± 3.05 | 2.34 ± 0.04 |

**Table 11: Results of PMP, $n = 500$**

| Methods | $p = 5$ | | $p = 10$ | | $p = 15$ | | $p = 20$ | |
|---|---|---|---|---|---|---|---|---|
| | Gap (%) | Time (s) | Gap (%) | Time (s) | Gap (%) | Time (s) | Gap (%) | Time (s) |
| Gurobi | 0.00 ± 0.00 | 13.93 ± 2.09 | 0.00 ± 0.00 | 12.94 ± 3.66 | 0.00 ± 0.00 | 13.26 ± 3.26 | 0.00 ± 0.00 | 15.83 ± 7.39 |
| Greedy-swap | 0.03 ± 0.06 | 6.98 ± 0.32 | 0.08 ± 0.16 | 42.38 ± 9.96 | 0.12 ± 0.16 | 90.33 ± 18.50 | 0.16 ± 0.26 | 158.10 ± 42.65 |
| Random-swap | 12.48 ± 3.20 | 0.11 ± 0.00 | 23.03 ± 4.86 | 0.24 ± 0.02 | 21.28 ± 3.80 | 0.36 ± 0.00 | 27.96 ± 5.83 | 0.50 ± 0.00 |
| SA | 6.04 ± 3.31 | 0.20 ± 0.02 | 9.84 ± 2.11 | 0.23 ± 0.03 | 9.77 ± 1.96 | 0.23 ± 0.01 | 11.94 ± 4.48 | 0.24 ± 0.00 |
| Maranzana | 13.02 ± 6.42 | 0.85 ± 0.02 | 21.53 ± 5.18 | 1.85 ± 0.32 | 32.53 ± 13.36 | 2.28 ± 0.38 | 49.62 ± 23.42 | 2.45 ± 0.38 |
| **PPO-swap** | 3.54 ± 1.81 | 0.34 ± 0.00 | 6.89 ± 3.09 | 0.64 ± 0.00 | 9.22 ± 2.26 | 0.95 ± 0.00 | 8.77 ± 2.33 | 1.25 ± 0.00 |
| **PPO-no-vor** | 5.17 ± 2.91 | 0.35 ± 0.02 | 10.38 ± 2.87 | 0.65 ± 0.04 | 13.07 ± 3.44 | 0.95 ± 0.07 | 12.79 ± 2.25 | 1.22 ± 0.02 |

| Methods | $p = 25$ | | $p = 30$ | | $p = 35$ | | $p = 40$ | |
|---|---|---|---|---|---|---|---|---|
| | Gap (%) | Time (s) | Gap (%) | Time (s) | Gap (%) | Time (s) | Gap (%) | Time (s) |
| Gurobi | 0.00 ± 0.00 | 12.24 ± 4.69 | 0.00 ± 0.00 | 12.88 ± 2.59 | 0.00 ± 0.00 | 15.51 ± 4.73 | 0.00 ± 0.00 | 10.85 ± 2.49 |
| Greedy-swap | 0.03 ± 0.06 | 244.44 ± 55.16 | 0.15 ± 0.14 | 327.94 ± 59.99 | 0.10 ± 0.18 | 455.31 ± 89.73 | 0.16 ± 0.19 | 553.84 ± 113.75 |
| Random-swap | 30.67 ± 7.03 | 0.65 ± 0.00 | 31.33 ± 7.01 | 0.84 ± 0.07 | 37.15 ± 9.91 | 0.96 ± 0.00 | 37.08 ± 6.86 | 1.13 ± 0.00 |
| SA | 14.98 ± 4.48 | 0.25 ± 0.01 | 16.65 ± 6.79 | 0.26 ± 0.00 | 21.02 ± 10.72 | 0.27 ± 0.01 | 32.36 ± 17.15 | 0.28 ± 0.01 |
| Maranzana | 62.48 ± 35.66 | 2.93 ± 0.57 | 77.45 ± 39.77 | 3.13 ± 0.57 | 97.59 ± 53.10 | 3.35 ± 0.50 | 119.68 ± 67.72 | 3.46 ± 0.71 |
| **PPO-swap** | 10.43 ± 2.47 | 1.56 ± 0.00 | 9.13 ± 2.40 | 1.87 ± 0.00 | 10.39 ± 2.80 | 2.14 ± 0.00 | 12.62 ± 2.61 | 2.43 ± 0.01 |
| **PPO-no-vor** | 13.58 ± 2.94 | 1.51 ± 0.01 | 14.12 ± 2.41 | 1.80 ± 0.03 | 14.27 ± 2.57 | 2.06 ± 0.04 | 14.84 ± 2.98 | 2.36 ± 0.05 |

**Table 12: Results of PMP, $n = 1000$**

| Methods | $p = 5$ | | $p = 10$ | | $p = 15$ | | $p = 20$ | |
|---|---|---|---|---|---|---|---|---|
| | Gap (%) | Time (s) | Gap (%) | Time (s) | Gap (%) | Time (s) | Gap (%) | Time (s) |
| Gurobi | 0.00 ± 0.00 | 148.46 ± 56.19 | 0.00 ± 0.00 | 138.54 ± 34.88 | 0.00 ± 0.00 | 139.57 ± 47.67 | 0.00 ± 0.00 | 124.04 ± 31.42 |
| Greedy-swap | 0.05 ± 0.14 | 37.32 ± 9.10 | 0.19 ± 0.23 | 148.49 ± 32.82 | 0.11 ± 0.16 | 335.14 ± 79.21 | 0.13 ± 0.13 | 577.60 ± 89.65 |
| Random-swap | 15.13 ± 5.75 | 0.17 ± 0.00 | 20.55 ± 4.65 | 0.35 ± 0.01 | 22.86 ± 5.26 | 0.55 ± 0.01 | 26.16 ± 3.47 | 0.76 ± 0.03 |
| SA | 5.86 ± 1.91 | 0.32 ± 0.05 | 10.21 ± 4.04 | 0.32 ± 0.02 | 11.02 ± 4.01 | 0.38 ± 0.08 | 17.66 ± 6.68 | 0.38 ± 0.06 |
| Maranzana | 14.38 ± 4.62 | 2.56 ± 0.76 | 27.24 ± 10.14 | 5.04 ± 1.41 | 39.07 ± 9.93 | 6.89 ± 1.63 | 50.53 ± 13.16 | 7.80 ± 1.51 |
| **PPO-swap** | 2.59 ± 1.39 | 0.37 ± 0.01 | 7.23 ± 2.41 | 0.70 ± 0.01 | 8.43 ± 1.79 | 1.04 ± 0.01 | 11.22 ± 1.81 | 1.37 ± 0.01 |
| **PPO-no-vor** | 6.62 ± 3.62 | 0.36 ± 0.01 | 9.17 ± 2.83 | 0.68 ± 0.02 | 13.58 ± 3.89 | 1.00 ± 0.02 | 15.19 ± 4.15 | 1.33 ± 0.02 |

| Methods | $p = 25$ | | $p = 30$ | | $p = 35$ | | $p = 40$ | |
|---|---|---|---|---|---|---|---|---|
| | Gap (%) | Time (s) | Gap (%) | Time (s) | Gap (%) | Time (s) | Gap (%) | Time (s) |
| Gurobi | 0.00 ± 0.00 | 113.52 ± 21.09 | 0.00 ± 0.00 | 120.08 ± 42.90 | 0.00 ± 0.00 | 110.02 ± 30.12 | 0.00 ± 0.00 | 117.00 ± 34.36 |
| Greedy-swap | 0.04 ± 0.06 | 894.52 ± 137.87 | 0.10 ± 0.14 | 1278.23 ± 131.81 | 0.12 ± 0.16 | 1183.25 ± 179.11 | 0.11 ± 0.13 | 1410.36 ± 257.62 |
| Random-swap | 29.48 ± 7.10 | 0.99 ± 0.02 | 32.57 ± 4.74 | 1.28 ± 0.01 | 34.80 ± 5.06 | 1.22 ± 0.03 | 34.20 ± 5.22 | 1.40 ± 0.03 |
| SA | 20.91 ± 4.47 | 0.41 ± 0.07 | 27.87 ± 12.02 | 0.46 ± 0.10 | 32.97 ± 10.51 | 0.34 ± 0.01 | 38.94 ± 20.99 | 0.35 ± 0.02 |
| Maranzana | 72.42 ± 23.02 | 8.68 ± 1.74 | 88.34 ± 29.81 | 8.42 ± 1.49 | 101.03 ± 38.90 | 10.00 ± 2.67 | 127.20 ± 48.11 | 10.78 ± 3.06 |
| **PPO-swap** | 11.31 ± 2.23 | 1.70 ± 0.01 | 12.91 ± 2.22 | 2.05 ± 0.02 | 14.24 ± 2.52 | 2.33 ± 0.01 | 13.39 ± 2.37 | 2.65 ± 0.02 |
| **PPO-no-vor** | 16.48 ± 4.43 | 1.65 ± 0.03 | 19.16 ± 3.86 | 1.98 ± 0.03 | 18.64 ± 3.71 | 2.26 ± 0.03 | 19.03 ± 2.61 | 2.57 ± 0.02 |

