# OpenReview forum: "Unified and Generalizable Reinforcement Learning for Facility Location Problems on Graphs"
_ACM.org/TheWebConf/2025/Conference — WWW 2025 Oral_

### Official Review · Reviewer_16fL · 2024-11-13

**Novelty:** 4
**Technical Quality:** 5

**Review:**

####Summery

This paper presents a novel approach for solving the facility location problem on graphs. In this paper, a graph neural network is utilized in a reinforcement learning framework to solve the facility relocation problem by iteratively selecting facilities to be removed and inserted. Subsequently, the process can be directly used to solve the p-median problem as a subroutine in the exchange framework. Extensive experiments on synthetic and real-world datasets have shown that the method proposed in this paper is capable of producing high-quality solutions on large graphs.

####Strengths

1.The paper proposes a unified framework that can simultaneously solve two types of facility location problems: the p-median problem and the facility relocation problem.

2.The paper integrates graph neural networks into a reinforcement learning framework that exploits the complex structural information of graphs to improve the generalization and representation capabilities of the model.

3.The model proposed in the paper demonstrates good scalability and achieves significant speedups on large-scale datasets compared to traditional solvers such as Gurobi.

####Weaknesses

1.Does the iterative approach used in the paper to pick removed facility and inserted facility tend to fall into local optimality?

2.I Would like more explanation of the model architecture diagram, e.g., what does the arrow pointing to “filter” mean in the diagram?

3.Why is the reward defined as such according to equation (6)? Has it been considered to be defined as the improvement ratio between the states before and after step i?

4.The method proposed in the paper has a great advantage over traditional methods in terms of time consumption, however, there remains scope for enhancement in terms of overall performance efficacy.

5.It would be better to add a comparison with other deep learning-based methods.

6.Overall writing needs to be more standardized, e.g., the first occurrence of abbreviations needs to be explained.

**Questions:**

1.Does the iterative approach used in the paper to pick removed facility and inserted facility tend to fall into local optimality?

2.I Would like more explanation of the model architecture diagram, e.g., what does the arrow pointing to “filter” mean in the diagram?

3.Why is the reward defined as such according to equation (6)? Has it been considered to be defined as the improvement ratio between the states before and after step i?

4.The method proposed in the paper has a great advantage over traditional methods in terms of time consumption, however, there remains scope for enhancement in terms of overall performance efficacy.

5.It would be better to add a comparison with other deep learning-based methods.

6.Overall writing needs to be more standardized, e.g., the first occurrence of abbreviations needs to be explained.

**Reviewer Confidence:**

3: The reviewer is confident but not certain that the evaluation is correct

**Scope:**

3: The work is somewhat relevant to the Web and to the track, and is of narrow interest to a sub-community

---

### Official Review · Reviewer_kmyi · 2024-12-02

**Novelty:** 5
**Technical Quality:** 5

**Review:**

The paper addresses the challenge of solving facility location problems (FLPs) on graphs, which are computationally difficult due to their NP-hard nature. While Mixed-Integer Programming (MIP) solvers like Gurobi can find optimal solutions, they struggle with large instances, making algorithm efficiency crucial, especially in emergency scenarios. Traditional machine learning methods have been limited to myopic constructive approaches and simple Euclidean spaces. In contrast, this paper proposes a novel approach using deep reinforcement learning (DRL) to handle FLPs on weighted graphs, taking into account complex graph structures. The authors claim their method achieves a balance between solution quality and running time, with superior efficiency and consistent performance. Their model, trained on small graphs, is scalable and generates high-quality solutions, achieving a speedup of over 2000 times compared to Gurobi on instances with 1000 nodes. The method's practical value is demonstrated through experiments on Shanghai road networks.

Pros:
* The paper presents a significant advancement in solving facility location problems on graphs using deep reinforcement learning. The claim of achieving a speedup of over 2000 times compared to Gurobi on large instances is impressive and suggests a high-quality solution.
* The paper is well-written and clearly outlines the problem, the proposed solution, and the key results. The use of deep reinforcement learning to tackle complex graph structures is clearly explained.

Cons:
* The significance of the work would be better supported by a broader range of experiments and real-world case studies beyond Shanghai road networks.
* The paper lacks sufficient detail on the experimental setup, model architecture, and training process, which could pose challenges for reproducibility. Providing more technical details and access to code and datasets would enhance the paper's reproducibility.

**Questions:**

* Are there specific types of graphs or facility location scenarios (e.g., different geographical regions, varying graph densities) that you believe would be particularly challenging or informative for your method?
* Beyond the Shanghai road networks, have you tested your method on other real-world datasets? If so, what were the results, and how do they compare to the Shanghai case?
* Could you please provide more detailed information on your implementation details?

**Reviewer Confidence:**

2: The reviewer is willing to defend the evaluation, but it is likely that the reviewer did not understand parts of the paper

**Scope:**

3: The work is somewhat relevant to the Web and to the track, and is of narrow interest to a sub-community

---

### Official Review · Reviewer_qrMx · 2024-12-03

**Novelty:** 3
**Technical Quality:** 3

**Review:**

This paper proposes a novel swap-based deep reinforcement learning (DRL) approach to solve facility location problems (FLPs) on weighted graphs, focusing on two key problem types: the p-median problem (PMP) and the facility relocation problem (FRP). The method emphasizes scalability, generalizability, and the ability to handle complex graph structures, addressing limitations of traditional methods and previous learning-based approaches. The authors highlight the superiority of their improving-style DRL paradigm over constructive methods commonly used in machine learning for FLPs. Experimental results on synthetic and real-world datasets, including Shanghai Road networks, demonstrate great efficiency and competitive solution quality.
Strengths:
1. The proposed method achieves good scalability and speedup, validated through experiments on Shanghai road networks.
2. The improving-style DRL paradigm demonstrates generalizability, outperforming some baselines in diverse scenarios.

Weaknesses:
1. The authors selected relatively few baselines and datasets, leaving a gap in understanding how the method fares against the latest techniques. In Table 1, apart from two variants of Algorithm 1, there are only three baselines. A similar issue is present in Table 2.
2. The paper lacks detailed ablation studies on the swap-based strategy and the specific reinforcement learning architecture, which are critical to understanding the contributions of individual components.
3. While the method scales well compared to Gurobi, the scalability on graphs larger than 1000 nodes or dynamic scenarios requiring frequent updates is not thoroughly addressed.
4. How did the authors consider the tradeoff between effectiveness and efficiency? In Table 1, the improvement ratio and the optimality gap show significantly worse performance compared to other baselines, and the improvement in running time is not very substantial. A similar issue is observed in Table 2. Additionally, running time reflects only a part of efficiency. The authors should discuss the advantages of the proposed method’s efficiency from more dimensions.

**Questions:**

See the weaknesses.

**Reviewer Confidence:**

3: The reviewer is confident but not certain that the evaluation is correct

**Scope:**

3: The work is somewhat relevant to the Web and to the track, and is of narrow interest to a sub-community

---

### Official Review · Reviewer_rwV9 · 2024-12-03

**Novelty:** 6
**Technical Quality:** 6

**Review:**

Overall, this is a well written paper that addresses the problem of facility placement on graphs, presenting a very competitive approach that empirically evaluates very well in terms of fidelity and computational efficiency.

Facility placement is somewhat limited in terms of impact within the graph domain, however, for its (slightly) niche applicability, the proposed solution is reasonable and evaluates very well. The authors might consider an application setting outside of road networks for more direct applicability within the 'web' domain. However, I understand that many graph methods--regardless of application--appear at this venue (note: justifying scope/applicability rating).

Strengths:

1. The paper is well written and the proposed method is well described. The choice of PPO is reasonable

2. The paper evaluates very well, especially on large graphs.

3. Solving the problem in a graph space allows for non-euclidean facility placement

Weaknesses:

1. As mentioned above, authors could consider applications outside of road networks. The applicability issue here is that often in social or digital networks, the edge augmentation problem is more relevant [1,2] below. This is because one is interested in increasing accessibility 'to'
a resource, where the resource is fixed within the network (e.g. jobs in a linkedin graph), where the utility is a form of link prediction for routing efficiency.

2. the voronoi-aware feature extractor seems to make specific spatial (euclidean) assumptions that might not be generalizable in other settings. Essentially, this is a coarsening filter that could introduce sensitivity depending on a choice of metric e.g. graph distance of some kind, euclidean distance.

This approach also doesn't seem to account for flow, e.g. in surveillance problems [3], being 'close' to a population even in graph distance is suboptimal to maximizing utility on edge flow. Applying the Voronoi approach necessarily coarsens the graph and may miss valuable nodes at a sub-segment level

Minor:

Some related works have focused on routing fairness for facility placement on graphs. This has applicability in road networks (with some demographic attributes). I don't expect these methods as a baseline, but group-fairness might be consideration for extending this work.

[1] https://dl.acm.org/doi/abs/10.1145/3461702.3462615
[2] https://dl.acm.org/doi/abs/10.1145/3593013.3594105
[3] https://dl.acm.org/doi/abs/10.1145/1281192.1281239

**Questions:**

Do you have sensitivity results with respect to the choice of metrics in the Voronoi tessellation? Is this choice even an issue for handling the surveillance use-case?

**Reviewer Confidence:**

4: The reviewer is certain that the evaluation is correct and very familiar with the relevant literature

**Scope:**

3: The work is somewhat relevant to the Web and to the track, and is of narrow interest to a sub-community

---

### Official Review · Reviewer_5sgY · 2024-12-04

**Novelty:** 5
**Technical Quality:** 5

**Review:**

This work is a substantial contribution to the field of facility location optimization, offering a novel and generalizable approach with demonstrated advantages in speed, scalability, and solution quality. With minor enhancements in real-world validation and simplification for broader audiences, it could serve as a benchmark in the domain.
pros:
1) The use of reinforcement learning for improving solutions rather than constructing them is novel and impactful.
2) Shows consistent performance across various metrics, including solution quality, computational efficiency, and adaptability.

cons:
1) While experiments on Shanghai road networks are impressive, more diverse real-world case studies would strengthen the paper's impact.
2) The computation of node features might become a bottleneck in scenarios requiring real-time responses for very large-scale instances.

**Questions:**

How does the model perform when the graph structures or demand distributions differ significantly from the training set? Could you provide additional examples or benchmarks that demonstrate its adaptability?

How sensitive are the model's performance and scalability to hyperparameters such as the number of GNN layers or the size of Voronoi-aware features? Could these choices impact generalizability?

**Reviewer Confidence:**

3: The reviewer is confident but not certain that the evaluation is correct

**Scope:**

3: The work is somewhat relevant to the Web and to the track, and is of narrow interest to a sub-community